# Exploring prevalence and factors associated with depression and anxiety symptoms among Bangladeshi graduates: a GIS-based cross-sectional study

## Research Article

mental health; unemployment; job seeking; prevalence; graduate students; spatial analysis

**Corresponding author:**
Mohammed A. Mamun;
Email: mamun@thechinta.org

Abdullah Al Habib[1,2] ⓘ, Imtiaz Uddin[1,5] ⓘ, Mehedee Hasan[1] ⓘ,
Firoj Al-Mamun[1,3,4] ⓘ, Moneerah Mohammad Almerab[6] ⓘ, David Gozal[7] ⓘ and
Mohammed A. Mamun[1,3,4] ⓘ

[1]CHINTA Research Bangladesh, Dhaka, Bangladesh; [2]Department of Government & Politics, Jahangirnagar University, Dhaka, Bangladesh; [3]Department of Public Health, University of South Asia, Dhaka, Bangladesh; [4]Department of Public Health and Informatics, Jahangirnagar University, Dhaka, Bangladesh; [5]Department of Sociology, University of Chittagong, Chattogram, Bangladesh; [6]Department of Psychology, College of Education and Human Development, Princess Nourah Bint Abdulrahman University, Riyadh, Saudi Arabia and [7]Department of Pediatrics, Joan C. Edwards School of Medicine, Marshall University, Huntington, WV, USA

## Abstract

Depression and anxiety are common mental health issues globally, yet limited research has focused on job seekers in Bangladesh. This study examines the prevalence and associated factors of depression and anxiety symptoms among Bangladeshi graduates seeking employment. A cross-sectional study was conducted among graduates from two public universities in Bangladesh, using face-to-face interviews and a semi-structured questionnaire. Data were collected between March and April 2024 through convenience sampling. Chi-square tests and logistic regression were used for analysis with SPSS software. Among the participants, 46.8% experienced depressive symptoms and 67.8% had anxiety symptoms, with 42.3% experiencing both. Factors associated with a reduced risk of depressive symptoms included being a first child (OR = 0.48, 95% CI: 0.25–0.93, $p$ = 0.031) and exam satisfaction (OR = 0.22, 95% CI: 0.12–0.39, $p$ < 0.001). Lower symptoms of anxiety were associated with being male (OR = 0.45, 95% CI: 0.25–0.80, $p$ = 0.007), first-born status (OR = 0.45, 95% CI: 0.22–0.92, $p$ = 0.030), financial contribution to family (OR = 0.40, 95% CI: 0.19–0.81, $p$ = 0.011), over 12 months of preparation (OR = 0.37, 95% CI: 0.15–0.92, $p$ = 0.034) and exam satisfaction (OR = 0.40, 95% CI: 0.22–0.71, $p$ = 0.002). Intentionally unemployed participants had a higher risk of anxiety symptoms (OR = 1.70, 95% CI: 1.00–2.89, $p$ = 0.046). This study reveals high rates of depressive and anxiety symptoms among job-seeking graduates in Bangladesh. Socio-demographic and job-related factors appear to significantly impact mental health, underscoring the need for a holistic approach to address these challenges. Targeted mental health interventions and increased public awareness are essential to support vulnerable groups in navigating the highly competitive job market.

## Impact Statement

The findings of this study hold important implications for stakeholders, including policymakers, educators, mental health professionals and the public. By uncovering the prevalence and predictors of depression and anxiety symptoms among job-seeking graduates in Bangladesh, this study emphasizes the urgent need for targeted mental health interventions for this previously unidentified vulnerable group. *Locally*, these insights can guide university administrators and career counselors in developing support systems specifically designed to help graduates transition into the workforce. Understanding the socio-demographic and job-related factors contributing to mental health challenges enables institutions to implement focused interventions, reduce stigma and promote well-being among students and alumni. *Regionally*, the findings support public health policies that prioritize mental health services for young adults, both within educational settings and through community programs. Policymakers can use this knowledge to allocate resources more effectively, addressing the mental health needs of job seekers and reducing the burden on this demographic. *Internationally*, the study adds to the understanding of mental health challenges faced by graduates in low- and middle-income countries, highlighting the interplay of socio-demographic factors, economic pressures and cultural expectations. This research highlights the importance of addressing mental health in the context of employment transitions, offering valuable insights that can contribute to better mental health outcomes and resilience among young adults entering the workforce.

## Introduction

The World Health Organization (WHO, 2024) identifies depression and anxiety as the most globally prevalent mental health disorders. Depression may manifest as a variety of debilitating symptoms, including sleep disturbances, appetite changes, feeling of hopelessness, thoughts of death, low self-esteem, fatigue and difficulty concentrating. Generalized anxiety disorder similarly impacts mental and physical health, with symptoms, such as chronic worry, difficulty managing uncertainty, restlessness, indecisiveness, fatigue, muscle tension and nausea (Ruscio et al., 2017). Large-scale studies provide insight into the prevalence of these conditions worldwide. For instance, a study conducted across 27 European countries involving 258,888 respondents reported the prevalence of depression at 6.38% (Arias-Arias et al., 2021). Similarly, a global study with 147,261 adults found that 3.7% of participants had experienced an anxiety disorder at some point in their lives (Ruscio et al., 2017).

Among adolescents, depression and anxiety are increasingly recognized as critical public health concerns. According to the WHO, approximately 15% of adolescents worldwide experience mental health disorders, with depression and anxiety ranking as the leading conditions in this age group (World Health Organization, 2021). It is reported that adolescence is a particularly vulnerable period, with rapid psychological, social and biological changes contributing to an increased risk of mental health disorders. Previous studies conducted among young adults have reported that depression and anxiety were highest among 18 to 29 years participants (Terlizzi and Zablotsky, 2024). A cohort study conducted in the US observed a significant increase in depression diagnoses among young individuals from 2017 to 2021, with a 60% rise in prevalence, while anxiety without depression also saw a 35.2% increase (Xiang et al., 2024). Several factors for depression and anxiety, such as being female, a history of depressive symptoms, negative life events, unemployed youth, duration of unemployment, never married, second- and third-time migrant and family-related stressful events contributed significantly to depression (World Health Organization, 2021).

Employment-related stressors, particularly job insecurity and unemployment, have been shown to exacerbate both depression and anxiety. Research highlights strong correlations between these mental health conditions and employment factors (McKee-Ryan et al., 2005; Elovainio et al., 2012; Mamun et al., 2020; Mokona et al., 2020). For example, a U.S.-based study among young adults (ages 18–26 years) observed that job insecurity during the COVID-19 pandemic led to increased anxiety and depressive symptoms (Ganson et al., 2021). Likewise, a study in Great Britain with 3,581 participants revealed that individuals facing limited job security were twofold more likely to experience depression (Meltzer et al., 2010). In low- and middle-income countries (LMIC), the situation is equally concerning. In a study conducted in Southern Ethiopia among unemployed youth, the prevalence of depression was 56.7%, where being male, experiencing long-term unemployment (≥ 1 years), low self-esteem, poor social support and current alcohol use were significantly associated with the symptoms of depression (Mokona et al., 2020). Similarly, in India, the comorbidity of depression and anxiety symptoms was reported, with 87% of depressed participants also suffering from anxiety disorder (Sahoo and Khess, 2010).

In recent years, the job market in Bangladesh has become increasingly competitive, as the growth in the number of college and university graduates outpaces the creation of new job opportunities in both government and private sectors (Apu, 2023; Hossen, 2023; The Daily Star, 2024). According to the Labor Force Survey by the Bangladesh Bureau of Statistics (BBS, 2023), approximately 800,000 graduates were unemployed in 2022. Between 2017 and 2022, the number of unemployed graduates doubled, with the unemployment rate rising from 11.2% in 2016–2017 to 12% in 2022 (BBS, 2023; Zaman, 2023). This increase in unemployed youth has led to fierce competition for available job positions, often resulting in frustration for well-prepared candidates who fail to secure employment in their desired fields (Roy, 2016; Islam and Amanullah, 2024). Lack of employment leads to adverse psychological or mental health consequences among the graduates, such as depression, stress, anxiety, suicidal ideation, insomnia, less problem-solving ability, and so on (Artazcoz et al., 2004; Reneflot and Evensen 2014; Cassidy and Wright, 2008; Lim et al., 2018; Maeda et al., 2019; Mæhlisen et al., 2018;). Moreover, unemployment brings feelings of frustration or of being neglected that might lead not only to mental health suffering but, in extreme cases, may develop into addiction to substances or criminal activity (Lim et al., 2018; Rahman, 2024). In Bangladesh, a study conducted among a relatively limited cohort of Bangladesh Civil Service Job Seekers in the quest for psychological conditions reported a prevalence of moderate to severe depression (49.3%) and anxiety (53.6%) symptoms (Rafi et al., 2019). Another study conducted among 1,066 unemployed youth in different cities in Bangladesh reported a very high prevalence rate of depression of 81.1% and anxiety of 61.5% symptoms (Mamun et al., 2020).

In light of the limited number of studies conducted on job seekers' mental health and factors related to these psychological health problems, the present study aimed to investigate previously unexplored job preparation-related factors to depression and anxiety. Moreover, this study represents a pioneering effort to provide nationwide, GIS-based insights into the prevalence of depression and anxiety symptoms. By identifying division-specific zones with higher prevalence rates, the research highlights the geographical disparities in mental health burdens across the country. By mapping these variations, the study aims to facilitate the development of more targeted and effective interventions, allowing policymakers and healthcare providers to address the magnitude of psychological issues with greater precision and efficacy.

## Methods

### *Study participants and procedure*

After completion of the university requirements in Bangladesh, a newly graduated candidate will begin the search for jobs, mostly public service-related employment opportunities and less often job-seeking efforts in the private sector (Emon, 2018).

A cross-sectional study was conducted among university graduates from two different public universities, Jahangirnagar University and Chittagong University in Bangladesh, who were preparing for jobs in government and private offices in Bangladesh. These universities were selected based on their diversity in student population, representing graduates from all districts in Bangladesh, which enhances the generalizability of the findings to a national context. Jahangirnagar University, located near the capital Dhaka, attracts students from both urban and rural settings, while Chittagong University, situated in the southeastern part of the country, includes students from coastal and remote areas. This diversity

ensures that the sample represents a wide spectrum of socio-demographic backgrounds.

A team of three members operated data collection via face-to-face interviews through a semi-structured questionnaire. The study was conducted between March and April 2024. A convenience sampling technique was used to collect data from respondents via a questionnaire. This method introduces selection bias, limiting the generalizability of the findings to the entire Bangladeshi population. However, efforts were made to mitigate this limitation by ensuring inclusiveness during data collection. Data were gathered in various locations, such as departments, student dormitories and university libraries, which are common meeting points for job-preparing graduates. Additionally, participants were drawn from diverse academic disciplines and year groups to capture a heterogeneous sample. In total, 600 questionnaires were distributed to the participants and around 20 min were required by the participants to answer the questions. Data were collected from 495 respondents with an 82.5% response rate. Due to inconsistency and missing information, 29 incomplete questionnaires were removed and 466 samples were retained for data analysis.

### Measures

#### Sociodemographic factors

This study included the following sociodemographic variables: gender (male vs. female), location (urban vs. rural), religion (Islam vs. Hindu & others), family type (nuclear vs. joint), number of family members (five or less vs. more than five), family income category (lower vs. middle vs. higher), birth order (first vs. second vs. third or more), relationship status (single vs. married), graduation year (2020 or before vs. 2021–2022 vs. 2023–2024), having a part-time job (yes vs. no) and contribution to family income (yes vs. no).

#### Health and behavioral variables

Preparation time category, targeted job, taking coaching, monthly expenses for preparation, preparatory exam satisfaction and being self-employed were collected as job preparation-related variables.

#### Mental health problems

Depressive symptoms were assessed using the Patient Health Questionnaire (PHQ-9) (Kroenke et al., 2001). Participants were instructed to respond based on their experiences over the past 2 weeks, with items including statements like "Little interest or pleasure in doing things." The PHQ-9 is a 9-item scale that utilizes a 4-point Likert scale, where responses range from 0 to 3 (Not at all = 0, Several days = 1, More than half the days = 2, Nearly every day = 3). The total score ranges from 0 to 27, with higher scores indicating greater depressive symptoms. A cut-off score of ≥10 was applied to identify significant depressive symptoms. The internal consistency of the PHQ-9 was measured using Cronbach's alpha coefficient as 0.87.

Symptoms of anxiety were assessed using the Generalized Anxiety Disorder (GAD-7) (Spitzer et al., 2006). Participants were asked to reflect on their experiences over the past 2 weeks, with items including statements, such as "Feeling nervous, anxious, or on edge." The GAD-7 is a 7-item scale that uses a similar 4-point Likert scale, where responses range from 0 to 3 (Not at all = 0, Several days = 1, More than half the days = 2, Nearly every day = 3). Scores on the GAD-7 range from 0 to 21, with higher scores indicating greater anxiety symptoms. A cut-off score of ≥5 was used to identify elevated anxiety levels. The internal consistency of the GAD-7 was measured using Cronbach's alpha coefficient as 0.82.

### Ethical consideration

This study adhered to the 2013 Helsinki Declaration and received ethical approval from CHINTA Research Bangladesh [ref: chinta/2023/12]. Informed written consent was obtained from all participants, who were assured of confidentiality and the voluntary nature of their involvement. Measures were taken to anonymize data and ensure privacy. Participants were also informed about available mental health support services, and it was emphasized that their participation would not impact their academic standing. The study upheld the principles of participant dignity, autonomy and well-being throughout the research process.

### Statistical analysis

After the data collection, the responses were recorded in Google Forms, which were then cleaned and prepared for final analysis by using Microsoft Excel 2021. Then, the Statistical Package for the Social Sciences (SPSS-25) was used to analyze the data. In the analysis, both descriptive statistics (frequency and percentages) and inferential statistics (chi-square and logistic regression) were used. The association between depressive symptoms, anxiety symptoms and the study variables was identified by using the chi-square test. The factors linked to anxiety and depressive symptoms were found through logistic regression. Results were reported from the adjusted model with their corresponding 95% confidence interval. The significance level for each statistical test was set at $p < 0.05$, with a 95% confidence interval. The GIS mapping was executed using the ArcGIS 10.8.2 software, which explored spatial distribution of depression and anxiety symptoms across divisions in Bangladesh. First, the geographic locational data of each respondent were matched by divisions and then distributed in maps as depressive and anxiety symptoms.

### Results

#### Description of the study participants

Around 60.7% of participants were female, 58.3% were from rural areas and 87% were Muslim. Most of the participants came from nuclear families (84%), had five or fewer family members (61.7%) and belonged to middle-income households (20,000–40,000 BDT) (40.5%). About 37.8% were firstborn, 95.2% were single and 50.1% graduated in 2023–2024. Over half of them (52.8%) had no part-time jobs, and 83.5% did not contribute financially to their families. Regarding job preparation efforts, 57% spent 0–6 months preparing, with 62.6% targeting Bangladesh Civil Service (BCS) jobs. Most participants had a first-class CGPA (95.8%), attended coaching sessions (71.8%) and spent under 2,000 BDT monthly on these sessions (64.8%). Additionally, 68.8% were dissatisfied with their preparatory exam results, and 51.6% were unemployed during their preparation period (Table 1).

#### Associations with the symptoms of depression and anxiety

Table 1 reports the association between socio-demographic information, job preparatory variables and symptoms of depression.

**Table 1.** Description of the variables and their associations with anxiety and depressive symptoms

| Variables | Total (n, %) | Anxiety symptoms (n = 316, 67.8%) | | Depressive symptoms (n = 218, 46.8%) | |
|---|---|---|---|---|---|
| | | Yes (n, %) | $\chi^2$ value (*p*-value) | Yes (n, %) | $\chi^2$ value (*p*-value) |
| **Socio-demographic information** | | | | | |
| Gender | | | | | |
| Male | 183 (39.3) | 113 (61.7) | **5.074 (0.024)** | 82 (44.8) | 0.471 (0.493) |
| Female | 283 (60.7) | 203 (71.7) | | 136 (48.1) | |
| Location | | | | | |
| Urban | 193 (41.7) | 137 (71) | 1.324 (0.250) | 102 (52.8) | **4.755 (0.029)** |
| Rural | 270 (58.3) | 178 (65.9) | | 115 (42.6) | |
| Religion | | | | | |
| Islam | 402 (87) | 269 (66.9) | 2.288 (0.130) | 187 (46.5) | 0.254 (0.614) |
| Hindu and others | 60 (13) | 46 (76.7) | | 30 (50) | |
| Family type | | | | | |
| Nuclear | 389 (84) | 267 (68.6) | 0.407 (0.524) | 183 (47) | 0.030 (0.862) |
| Joint | 74 (16) | 48 (64.9) | | 34 (45.9) | |
| Number of family members | | | | | |
| Five or less | 284 (61.7) | 190 (66.9) | 0.445 (0.505) | 122 (43) | **4.264 (0.039)** |
| More than five | 176 (38.3) | 123 (69.9) | | 93 (52.8) | |
| Family income category | | | | | |
| Lower income | 125 (29.6) | 77 (61.1) | 4.806 (0.090) | 49 (39.2) | 5.138 (0.077) |
| Middle income | 171 (40.5) | 121 (70.8) | | 83 (48.5) | |
| Higher income | 126 (29.9) | 93 (73.8) | | 67 (53.2) | |
| Birth order | | | | | |
| First | 174 (37.8) | 111 (63.8) | 2.756 (0.252) | 72 (41.4) | **7.290 (0.026)** |
| Second | 141 (30.7) | 97 (68.8) | | 62 (44) | |
| Third or more | 145 (31.5) | 105 (72.4) | | 81 (55.9) | |
| Relationship status | | | | | |
| Single | 440 (95.2) | 304 (69.1) | **5.376 (0.020)** | 207 (47) | 0.021 (0.884) |
| Married | 22 (4.8) | 10 (45.5) | | 10 (45.5) | |
| Graduation year | | | | | |
| 2020 or before | 37 (8.1) | 23 (62.2) | 5.793 (0.055) | 17 (45.9) | 1.097 (0.578) |
| 2021–2022 | 191 (41.8) | 141 (73.8) | | 94 (49.2) | |
| 2023–2024 | 229 (50.1) | 145 (63.3) | | 101 (44.1) | |
| Having part-time job | | | | | |
| Yes | 220 (47.2) | 156 (70.9) | 1.832 (0.176) | 109 (49.5) | 1.279 (0.258) |
| No | 246 (52.8) | 160 (65) | | 109 (44.3) | |
| Contribution in family (money) | | | | | |
| Yes | 77 (16.5) | 45 (58.4) | 3.710 (0.054) | 37 (48.1) | 0.060 (0.807) |
| No | 389 (83.5) | 271 (69.7) | | 181 (46.5) | |
| **Job preparation-related variables** | | | | | |
| Preparation time category | | | | | |
| 0 to 6 months | 240 (57) | 163 (68.2) | 2.668 (0.263) | 111 (46.4) | 1.028 (0.598) |
| 6 to 12 months | 119 (28.3) | 76 (63.9) | | 52 (43.7) | |
| More than 12 months | 62 (14.7) | 47 (75.8) | | 32 (51.6) | |

(*Continued*)

**Table 1.** (*Continued*)

| Variables | Total (n, %) | Anxiety symptoms (n = 316, 67.8%) | | Depressive symptoms (n = 218, 46.8%) | |
|---|---|---|---|---|---|
| | | Yes (n, %) | $\chi^2$ value (*p*-value) | Yes (n, %) | $\chi^2$ value (*p*-value) |
| Targeted job | | | | | |
| BCS | 289 (62.6) | 202 (69.9) | 1.629 (0.202) | 129 (44.6) | 1.687 (0.194) |
| Other Gov't and private job | 173 (37.4) | 111 (64.2) | | 88 (50.9) | |
| Taking coaching | | | | | |
| Yes | 130 (28.2) | 91 (70) | 0.446 (0.504) | 60 (46.2) | 0.061 (0.805) |
| No | 331 (71.8) | 221 (66.8) | | 157 (47.4) | |
| Monthly expenses for preparation | | | | | |
| Less than 5,000 | 328 (75.1) | 198 (70) | 0.278 (0.598) | 138 (48.8) | 2.106 (0.147) |
| More than 5,000 | 109 (24.9) | 103 (66.9) | | 68 (44.2) | |
| Preparatory exam satisfaction | | | | | |
| Yes | 128 (31.2) | 77 (60.2) | **7.259 (0.007)** | 35 (27.3) | **31.225 (<0.001)** |
| No | 282 (68.8) | 207 (73.4) | | 161 (57.1) | |
| Being self-unemployed | | | | | |
| Yes | 232 (51.6) | 167 (72) | 3.124 (0.077) | 111 (47.8) | 0.103 (0.748) |
| No | 218 (48.4) | 140 (64.2) | | 101 (46.3) | |

Participants living in urban areas had a higher rate of depressive symptoms compared to rural areas ($\chi^2$ = 4.755, *p* = 0.029). The number of family members was associated with the symptoms of depression with more than five members in the family exhibiting a higher rate of depressive symptoms ($\chi^2$ = 4.264, *p* = 0.039). Furthermore, students with no preparatory exam satisfaction were significantly more likely to report the symptoms of depression ($\chi^2$ = 31.225, *p* < 0.001) compared to students with preparatory exam satisfaction.

Table 1 also shows socio-demographic information, job preparation related variables related to the symptoms of anxiety. Female exhibited more anxiety symptoms compared to males ($\chi^2$ = 5.074, *p* = 0.024). In addition, the prevalence of anxiety symptoms was significantly higher among single compared to married participants ($\chi^2$ = 5.376, *p* = 0.020). Moreover, students who reported being dissatisfied with their preparatory exam were significantly more prone to anxiety symptoms (73.4%; $\chi^2$ = 7.259, *p* = 0.007).

### Factors associated with symptoms of depression and anxiety

Based on Table 2, the significant variables associated with anxiety symptoms among job seekers included gender, birth order, contribution to family, preparation time, exam satisfaction and self-employment status. Male participants were less likely to experience anxiety symptoms compared to females (OR = 0.451, 95% CI: 0.252–0.804, *p* = 0.007). Firstborn individuals also had a lower risk of anxiety symptoms compared to those born third or later (OR = 0.454, 95% CI: 0.223–0.925, *p* = 0.030). Those who contributed financially to their families had a reduced risk of anxiety symptoms (OR = 0.401, 95% CI: 0.199–0.811, *p* = 0.011). Longer preparation time (>12 months) was associated with a lower risk of anxiety symptoms (OR = 0.375, 95% CI: 0.151–0.927, *p* = 0.034). Participants dissatisfied with their preparatory exam results had a higher likelihood of experiencing symptoms of anxiety (OR = 0.403, 95% CI: 0.227–0.715, *p* = 0.002). Finally, those who were intentionally

unemployed had an increased risk of anxiety symptoms (OR = 1.709, 95% CI: 1.009–2.892, *p* = 0.046).

Table 3 shows the factors associated with socio-demographic information, job preparation-related variables, and depression among job seekers. Birth order was significant, with firstborn individuals showing a lower risk of depressive symptoms compared to those who were third-born or later (OR = 0.487, 95% CI: 0.253–0.936, *p* = 0.031). Second, exam satisfaction was a significant factor; those dissatisfied with their preparatory exam results had a higher risk of depressive symptoms (OR = 0.225, 95% CI: 0.127–0.398, *p* < 0.001).

### Mental health symptoms across districts

As illustrated in Figure 1, GIS analysis of depression and anxiety symptoms across Bangladesh reveals significant variation in prevalence across regions (termed as divisions). However, for both symptoms of depression ($\chi^2$ = 11.552, *p* = 0.116) and anxiety ($\chi^2$ = 8,212, *p* = 0.314), we did not find significant associations with regional divisions. The highest prevalence of depressive symptoms was observed in Dhaka (55.9%), followed closely by Chattogram at 52.3%. The next tier of high prevalence included Barisal (47.8%), Rangpur (45.5%) and Khulna (42.9%). The regions with the lowest depressive symptom rates were Sylhet (37.5%), Mymensingh (35.1%) and Rajshahi (32.7%). In terms of anxiety symptoms, the highest prevalence was found in Chattogram, where 77.1% of participants reported anxiety symptoms, followed by Dhaka at 71.2%. Rangpur (66.7%), Barisal (65.2%) and Mymensingh (64.9%) also showed high levels of anxiety symptoms. Lower rates of anxiety symptoms were observed in Sylhet (62.5%), Khulna (63.3%) and Rajshahi (57.1%). Overall, participants from Chattogram and Dhaka were the most severely affected by both the symptoms of depression and anxiety, highlighting these highly populated regions as potential areas for targeted mental health interventions in Bangladesh.

**Table 2.** Factors associated with symptoms of anxiety in job-seeking graduates in Bangladesh

| Variable name | B | S.E. | Wald | Sig. | Exp(B) | 95% CI for EXP(B) | |
|---|---|---|---|---|---|---|---|
| | | | | | | Lower | Upper |
| Gender (Male) [Ref: Female] | −.797 | .296 | 7.269 | **.007** | .451 | .252 | .804 |
| Age | .104 | .128 | .655 | .418 | 1.110 | .863 | 1.427 |
| Location (Urban) [Ref: Rural] | .319 | .292 | 1.189 | .276 | 1.376 | .775 | 2.440 |
| Religion (Islam) [Ref: Hindu and others] | −.445 | .406 | 1.205 | .272 | .641 | .289 | 1.419 |
| Family type (Nuclear) [Ref: Joint] | −.113 | .423 | .071 | .789 | .893 | .390 | 2.048 |
| Number of Family Members [Ref: ≥ 6] | .114 | .324 | .123 | .726 | 1.121 | .593 | 2.116 |
| Birth Order | | | 6.436 | .040 | | | |
| First child [Ref: ≥ 3] | −.789 | .363 | 4.736 | **.030** | .454 | .223 | .925 |
| Second child [Ref: ≥ 3] | −.114 | .357 | .103 | .749 | .892 | .443 | 1.795 |
| Family Income | | | 1.267 | .531 | | | |
| Low income [Ref: High income] | −.393 | .359 | 1.204 | .272 | .675 | .334 | 1.362 |
| Middle Income [Ref: High income] | −.144 | .328 | .193 | .660 | .866 | .455 | 1.646 |
| Relationship Status (Unmarried) [Ref: Married] | .872 | .655 | 1.771 | .183 | 2.391 | .662 | 8.635 |
| Graduation Year | | | 1.300 | .522 | | | |
| 2021–2022 [Ref: 2020 or before] | .392 | .572 | .469 | .493 | 1.479 | .482 | 4.536 |
| 2023–2024 [Ref: 2020 or before] | .332 | .303 | 1.197 | .274 | 1.393 | .769 | 2.524 |
| Having Part-time Job [Ref: No] | .405 | .274 | 2.183 | .140 | 1.499 | .876 | 2.566 |
| Contribution to Family [Ref: No] | −.913 | .359 | 6.472 | **.011** | .401 | .199 | .811 |
| Preparation Time Category | | | 4.529 | .104 | | | |
| 6 to 12 months [Ref: 0 to 6 months] | −.729 | .476 | 2.346 | .126 | .482 | .190 | 1.226 |
| > 12 months [Ref: 0 to 6 months] | −.982 | .462 | 4.507 | **.034** | .375 | .151 | .927 |
| Targeted Job (BCS) [Ref: Other Gov't and Private job] | −.012 | .288 | .002 | .967 | .988 | .562 | 1.738 |
| Taking Coaching [Ref: No] | .140 | .334 | .176 | .675 | 1.150 | .598 | 2.212 |
| Monthly expenses (<5,000) [Ref: >5,000] | .271 | .309 | .768 | .381 | 1.311 | .716 | 2.401 |
| Satisfaction of Preparatory Exam [Ref: No] | −.910 | .293 | 9.663 | **.002** | .403 | .227 | .715 |
| Being Self-unemployed [Ref: No] | .536 | .268 | 3.981 | **.046** | 1.709 | 1.009 | 2.892 |
| Constant | −1.129 | 3.433 | .108 | .742 | .323 | | |

Nagelkerke $R^2$ = **18.1%**

## Discussion

The objectives of this study were to examine the prevalence of both depressive and anxiety symptoms and identify associated factors among job-seeking graduates in Bangladesh. Major findings were the inordinately elevated prevalence of such problems in this clearly vulnerable sector of the population. Notably, 42.3% of responders reported both the symptoms of depression and anxiety, while only 27.7% showed no symptoms of either condition. Key factors associated with the symptoms of depressive symptoms included birth order and dissatisfaction with preparatory exams, while for anxiety symptoms, being female, birth order, lack of family contribution, shorter preparation duration (0–6 months), dissatisfaction with preparatory exams and intentional unemployment emerged as significant risk factors.

To place our findings in the context of previous studies in Bangladesh, the rate of depressive symptoms recorded by the present study is similar to the one reported among Bangladeshi Civil Service job seekers (Rafi et al., 2019) but is considerably lower (81.1%) than the unemployed youth (Mamun et al., 2020). In contrast, prevalence of anxiety symptoms is higher than that reported for civil service job seekers (53.6%) and slightly exceeds the prevalence found among unemployed youth (Rafi et al., 2019; Mamun et al., 2020). Furthermore, the prevalence of depressive symptoms is significantly lower than the 80.2% found among Bangladeshi medical students (Biswas et al., 2021), highlighting potential differences in mental health challenges based on academic or career stress levels. In the broader South Asian context, findings from Kolkata, India, show comparable rates, with 54.4% for depressive symptoms and 61.8% for anxiety symptoms among highly educated migrant youth (Biswas et al., 2024). In Sri Lanka, however, rates are considerably lower, with 36% for depressive symptoms and 28% for anxiety symptoms among adolescent students (Rodrigo et al., 2010), possibly reflecting differences in socioeconomic pressures or support systems. Globally, the current study's depressive symptoms rate aligns with previous findings suggesting the rate of depressive symptoms was 56.7% among unemployed

**Table 3.** Factors associated with symptoms of depression in job-seeking graduates in Bangladesh

| Variable name | B | S.E. | Wald | Sig. | Exp(B) | 95% CI for EXP(B) | |
|---|---|---|---|---|---|---|---|
| | | | | | | Nagelkerke R$^2$ = 18.5% | |
| | | | | | | Lower | Upper |
| Gender (Male) [Ref: Female] | .027 | .275 | .009 | .923 | 1.027 | .599 | 1.759 |
| Age | .125 | .121 | 1.071 | .301 | 1.133 | .894 | 1.436 |
| Location (Urban) [Ref: Rural] | .309 | .261 | 1.394 | .238 | 1.362 | .816 | 2.273 |
| Religion (Islam) [Ref: Hindu and others] | −.227 | .348 | .426 | .514 | .797 | .402 | 1.577 |
| Family type (Nuclear) [Ref: Joint] | .374 | .382 | .961 | .327 | 1.454 | .688 | 3.072 |
| Number of Family Members [Ref: ≥ 6] | .152 | .297 | .260 | .610 | 1.164 | .650 | 2.085 |
| Birth Order | | | 4.832 | .089 | | | |
| First child [Ref: ≥ 3] | −.720 | .334 | 4.653 | **.031** | .487 | .253 | .936 |
| Second child [Ref: ≥ 3] | −.538 | .324 | 2.762 | .097 | .584 | .309 | 1.101 |
| Family Income | | | .973 | .615 | | | |
| Low income [Ref: High income] | −.310 | .337 | .844 | .358 | .734 | .379 | 1.421 |
| Middle Income [Ref: High income] | −.242 | .295 | .673 | .412 | .785 | .440 | 1.400 |
| Relationship Status (Unmarried) [Ref: Married] | .191 | .621 | .095 | .758 | 1.211 | .358 | 4.092 |
| Graduation Year | | | 1.937 | .380 | | | |
| 2021–2022 [Ref: 2020 or before] | −.128 | .526 | .059 | .808 | .880 | .314 | 2.466 |
| 2023–2024 [Ref: 2020 or before] | −.389 | .286 | 1.856 | .173 | .678 | .387 | 1.186 |
| Having Part-time Job [Ref: No] | .279 | .253 | 1.215 | .270 | 1.322 | .805 | 2.172 |
| Contribution to Family [Ref: No] | −.136 | .333 | .166 | .684 | .873 | .454 | 1.677 |
| Preparation Time Category | | | 2.200 | .333 | | | |
| 6 to 12 months [Ref: 0 to 6 months] | −.597 | .417 | 2.055 | .152 | .550 | .243 | 1.245 |
| > 12 months [Ref: 0 to 6 months] | −.307 | .403 | .578 | .447 | .736 | .334 | 1.622 |
| Targeted Job (BCS) [Ref: Other Gov't and Private job] | −.455 | .269 | 2.855 | .091 | .634 | .374 | 1.076 |
| Taking Coaching [Ref: No] | .083 | .309 | .073 | .787 | 1.087 | .594 | 1.990 |
| Monthly expenses (<5,000) [Ref: >5,000] | .475 | .285 | 2.781 | .095 | 1.607 | .920 | 2.808 |
| Satisfaction of Preparatory Exam [Ref: No] | −1.492 | .291 | 26.358 | **<.001** | .225 | .127 | .398 |
| Being Self-unemployed [Ref: No] | .201 | .249 | .657 | .418 | 1.223 | .751 | 1.991 |
| Constant | −2.267 | 3.215 | .497 | .481 | .104 | | |

Ethiopian (Mokona et al., 2020) and 39.5% among Korean job seekers; (Lim et al., 2018). However, it is notably higher than the rates reported in Western countries. For example, in the U.S., symptoms of depression and anxiety rates among the unemployed stand at 29% and 31%, respectively (Howe et al., 2012), while in Spain, rates are 51.5% for depressive symptoms and 35.5% for anxiety symptoms (Navarro-Abal et al., 2018). In Greece, during the post-financial crisis period, rates of 32.2% for the symptoms of depression and 39.7% for were reported for anxiety symptoms (Kokaliari, 2016), which are still lower than the figures observed in the present study. Thus, the prevalence of the symptoms of depression and anxiety in the current study is substantial and is particularly elevated compared to Western countries, likely reflecting unique socioeconomic challenges faced by job-seeking graduates in Bangladesh.

In this study, being the third-born or subsequent in birth order was associated with higher risks of both depression and anxiety symptoms among job seekers compared to first- and second-born children. This finding aligns with the research by Gates et al. (1988), who showed that firstborns tend to have significantly lower levels of depressive and anxiety symptoms than those born in subsequent order. Conversely, a study by Fukuya et al. (2021) reported that last-born children were less likely to experience mental health issues and exhibited more prosocial behaviors than first- or second-borns. The current study findings may be reflective of the unique socioeconomic and cultural context in Bangladeshi society, where elder children are often raised to assume familial responsibilities and benefit from mentorship from older family members, potentially making them more resilient and psychologically stable. In contrast, higher-order born children may receive more attention and be held less accountable, which could limit their exposure to challenging situations that build coping skills, making them more vulnerable to mental health issues in stressful contexts such as job seeking.

Being female emerged as a significant risk factor for anxiety symptoms among job-seeking graduates, a finding that concurs with established gender differences in mental health. A systematic

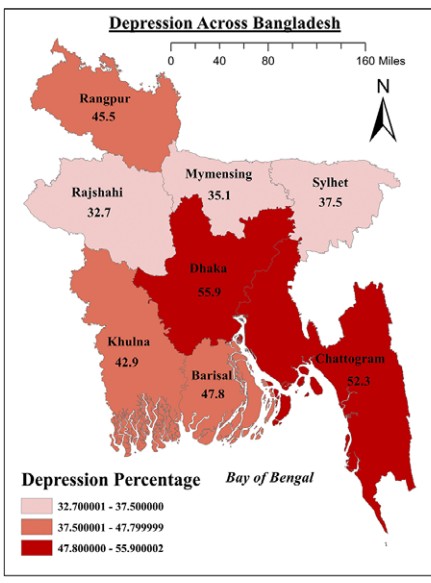
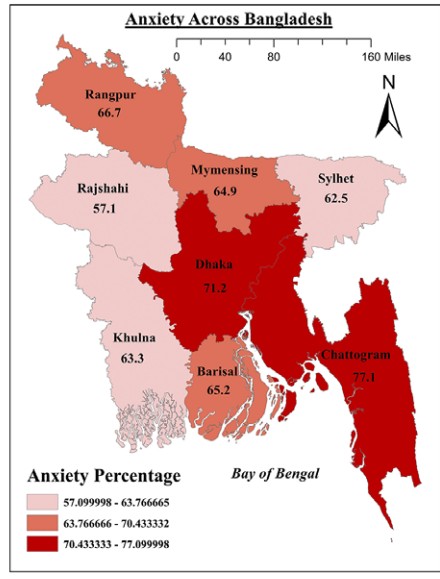

**Figure 1.** Depression and anxiety symptoms across divisions among job-seeking graduates in Bangladesh.

review and meta-analysis of studies conducted in Bangladesh highlighted that female-participants are at greater risk for both anxiety and depressive symptoms (Hosen et al., 2021). Similar observations have been reported in studies conducted in various cultural and geographical settings, whereby females consistently exhibited higher levels of anxiety than their male counterparts (Özdin and Bayrak Özdin, 2020; Maatouk et al., 2021). Several potential explanations have been proposed for this gender disparity in mental health outcomes. Biological factors, particularly hormonal differences, are believed to play a critical role. Sex hormones, such as estrogen and progesterone, influence various biological, behavioral and cognitive processes that may contribute to heightened vulnerability to anxiety and stress-related disorders in females. Besides, hormonal fluctuations during different life stages, such as menstruation, pregnancy and menopause, are known to affect mood regulation, potentially exacerbating anxiety symptoms (Li and Graham, 2017). Thus, a combination of biological, psychological and possibly social factors likely contributes to the higher prevalence of anxiety symptoms observed among female job seekers.

Dissatisfaction with job preparatory exams significantly contributed to the symptoms of depression and anxiety among job seekers. Mental health issues, including depression and anxiety, may arise when candidates are displeased and disappointed with their mock exam performances, which disrupts the development of a positive mindset for the actual test. Indeed, a previous study on university entrance test takers found that dissatisfied candidates were 2.66 times more likely to experience burnout (Mamun et al., 2021). Similarly, Nahrin et al. (2023) reported that dissatisfaction with mock exams could even lead to suicidal tendencies, alongside depression and anxiety, particularly among repeat test takers. The role of preparatory exams is crucial in instilling confidence, and when expectations are not met, it can heighten psychological distress. Furthermore, this study found that minimal preparation time (0 to 6 months) also contributes to anxiety symptoms among job seekers. Consistent with this, a quasi-experimental study demonstrated that adequate preparation can reduce test anxiety and enhance performance (Yusefzadeh et al., 2019). For this reason, in job-related preparation, a candidate must have enough time and have access to appropriate study strategies to prepare for upcoming

tests. Failure to accomplish these goals will otherwise generate anxiety symptoms about the test due to suboptimal preparation (Badrian et al., 2022).

In the present study, not being able to contribute financially to the family emerged as a significant factor for developing anxiety symptoms among job seekers. In Bangladesh, where many graduates feel a core responsibility to support their elderly parents, this inability to contribute can lead to heightened stress and anxiety. Similar findings were reported in Canada, where educators who were responsible for the care of older adults exhibited significantly higher levels of anxiety (Spadafora et al., 2022). Contributing to family needs, whether financially or through other forms of support, will enhance self-satisfaction and well-being (Kim and Sok, 2012). Furthermore, intentional unemployment, or self-unemployment, was identified as a significant factor in developing anxiety symptoms in this study, with unemployed participants showing nearly double the risk of anxiety symptoms. This finding aligns with previous research, where financial threat and hardship were found to be positively correlated with anxiety, depression and stress, while financial well-being was negatively correlated with anxiety (Mamun et al., 2020). The psychological impact of unemployment can lead to feelings of neglect and frustration, which may escalate to suicidal thoughts in extreme cases (Lim et al., 2018). These findings highlight the importance of financial stability and family support for mental well-being among job-seeking graduates.

The geographic locational data found that the two major divisions, namely Chattogram and Dhaka, were disproportionately affected by the symptoms of depression and anxiety while rural locations were less affected. Previous findings related to depressive and anxiety symptoms in urban and rural areas in Canada suggested that the risk of mental health problems increases in urban life due to reduced sense of community belonging (Romans et al., 2011). Another study in Korea reported heightened depression in urban participants later in life (Kim et al., 2004). Two other regions, Rangpur and Barisal, were also markedly affected and coincide with higher rates of poverty, a finding that resonates with data from the World Bank Group (World Bank, 2014). Poverty impacts a person's mental health (Lund et al., 2010; Ridley et al., 2020) and a

longitudinal study reported that family poverty from early life to adolescent period was the most significant factor for depression and anxiety (Najman et al., 2010). Indeed, child exposure to poverty increased the risk of facing depression and anxiety issues at 14- and 21-year follow-up (Najman et al., 2010).

While this study offers valuable insights into depression and anxiety symptoms among job seekers in Bangladesh, it has several limitations. First, the cross-sectional design limits the ability to establish causal relationships between risk factors and mental health outcomes; future research using longitudinal designs will be needed to explore the temporal dynamics of these associations. Besides, this study did not account for variables such as participants' mental health history, which could influence the recurrence of depression and anxiety, a potential area for further research. The study was also limited to graduates from two universities in Bangladesh (University of Chittagong and Jahangirnagar University) and employed a convenience sampling approach. This, along with a modest sample size, may restrict the generalizability of the findings. To build a more comprehensive understanding of mental health among job seekers in Bangladesh, future studies should include larger, more diverse samples and consider a nationwide scope.

## Conclusions and recommendations

This study highlights the high prevalence of the symptoms of depression and anxiety among job-seeking graduates in Bangladesh and identifies several socio-demographic and job-related factors associated with these mental health challenges. Our findings underscore that being female, having a higher birth order, lack of family financial contribution, dissatisfaction with preparatory exams and limited preparation time are significant factors of anxiety symptoms, while dissatisfaction with exams and higher birth order are also linked to symptoms of depression. These insights point to the need for a holistic approach to address mental health issues among job seekers, emphasizing personalized support and targeted mental health services for vulnerable groups.

To address the challenges faced by job-seeking graduates, we recommend several actionable interventions. Universities and career centers should establish integrated mental health support services, including psychotherapy, cognitive-behavioral therapy and stress management workshops. Structured exam preparation programs offering high-quality study resources and flexible schedules can alleviate exam-related anxiety. Implementing peer support networks and mentorship programs can foster emotional well-being and enhance coping mechanisms. Public awareness campaigns utilizing media and educational platforms should aim to reduce the stigma around mental health, encouraging individuals to seek professional help. Partnerships between government agencies, public health organizations and educational institutions can facilitate inclusive mental health programs, including financial aid for job seekers, mental health education and workplace well-being policies. Incorporating career counseling services and skills development workshops can better prepare graduates for the job market, easing employment-related stress. By implementing these recommendations, policymakers and stakeholders can create an environment that prioritizes mental health and well-being, leading to a healthier and more resilient workforce.

**Open peer review.** To view the open peer review materials for this article, please visit http://doi.org/10.1017/gmh.2025.21.

**Data availability statement.** The datasets will be made available to appropriate academic parties upon request from the corresponding author.

**Acknowledgements.** The authors thank all the participants and team members involved in the project.

**Author contributions.** This study was conceptualized by AAH and IU. The project was implemented and managed, including data collection to data entry, by AAH, IU and MH with direct support from MAM and FAM. It is worth noting that AAH and IU completed the data analysis using the SPSS, which were reviewed and finalized by FAM and MAM and validated by other authors. The project was directly supervised by FAM and MAM, as well as subsequently by MMA and DG. The initial draft of this study was written by AAH, whereas subsequent contributions were made by IU and MAM. All authors contributed to extensive edits and reviews. The final version is reviewed and approved by all authors.

**Financial support.** M.M.A. acknowledges the funding support currently being received from Princess Nourah bint Abdulrahman University Researchers Supporting Project Number (PNURSP2025R563), Princess Nourah bint Abdulrahman University, Riyadh, Saudi Arabia.

**Competing interest.** The authors declare none.

**Ethical standard.** This study adhered to the 2013 Helsinki Declaration and received ethical approval from CHINTA Research Bangladesh [ref: chinta/2023/12]. Informed consent was obtained from all participants, who were assured of confidentiality and the voluntary nature of their involvement. Measures were taken to anonymize data and ensure privacy. Participants were also informed about available mental health support services, and it was emphasized that their participation would not impact their academic standing. The study upheld the principles of participant dignity, autonomy and well-being throughout the research process.

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
