## [Reviewer Report]

I thank the authors for their contribution to the study of mental health of Bangladeshi graduates, which is a complete study with reliable conclusions. By scrutinizing this manuscript, I offer the following comments for the author’s consideration:

Introduction:

1. In the background section, it is recommended to describe the prevalence, risk factors, characteristics, and global trends of depression and anxiety in the adolescent population.

2. It is recommended to add relevant results from low- and middle-income countries on the topic of this study.

Methods.

1. The representativeness of the sampling method needs to be explained. how was sampling done in each school?

2. Socio-demographic factors need to be added for specific categorization.

Discussion

1. Some actionable recommendations should be made.

---

## [Reviewer Report]

The following comments may provide a more contextual approach to the manuscript:

Methods:

- How were the two universities selected? Please provide justification if they were selected conveniently. For example, are they the two biggest universities with the highest undergraduate enrollment in Bangladesh?

- Convenience sampling was used, introducing major selection bias, and therefore the results cannot be generalised to the Bangladeshi population.

- Depression and anxiety are clinical diagnoses which cannot be concluded based on a questionnaire alone. A more suitable outcome to be reported by this study are the presence of depressive and anxiety symptoms.

- Please explain the role of GIS in this study. It is good to have a subsection on its own to describe it. Is there any spatial data included as the independent variables?

Results:

- Please include 95% Confidence Interval every time (including in abstract) reporting the odds ratio.

- Subsection 3.4 was mistakenly labelled as “Mental Health Literacy Across District”. It is actually explaining the distribution of mental health symptoms across districts.

Discussion:

- The results should be discussed within the context of the sampling population and the main study outcome, i.e., depressive symptoms and anxiety symptoms, and not depression and anxiety.